# Adherence to Therapy in Glaucoma Treatment—A Review

**DOI:** 10.3390/jpm12040514

**Published:** 2022-03-22

**Authors:** Alexandra-Cătălina Zaharia, Otilia-Maria Dumitrescu, Mădălina Radu, Roxana-Elena Rogoz

**Affiliations:** Department of Ophthalmology, “Dr. Carol Davila” Central Military Emergency Hospital, 010825 Bucharest, Romania; otiliamariadumitrescu@gmail.com (O.-M.D.); madalina.rd@yahoo.com (M.R.); rogozroxana@yahoo.com (R.-E.R.)

**Keywords:** patient adherence, glaucoma, personalized care, patient compliance

## Abstract

Glaucoma is a chronic disease and the second leading cause of irreversible vision loss worldwide, whose initial treatment consists of self-administered topical ocular hypotensive eyedrops. Adherence with glaucoma medications is a fundamental problem in the care of glaucoma patients as up to 50% of patients fail to receive the intended benefits of the treatment. The literature has identified many barriers to patients’ compliance, from factors depending on the type of medication administered, communication between physician and patients, to factors dependent on patients’ behaviour and lifestyle. Failure to take medication as prescribed increases the risk that patients will not receive the desired benefit, which often leads to a worsening of the disease. Our aim is to synthesize the methods used for measuring adherence of patients to glaucoma therapy and the interventions used for addressing adherence, laying emphasis on a patient-centred approach, taking time to educate patients about their chronic disease and to assess their views on treatment.

## 1. Introduction

Glaucoma is an optic neuropathy that poses a significant public health problem, affecting approximatively sixty million people worldwide. The goal of the currently available glaucoma medication is to preserve visual function by lowering intraocular pressure to a level that is likely to prevent further optic nerve damage, but patients are unlikely to experience the clinical benefits of their therapy if they do not adhere to their medication regimen [1,2].

Poor adherence to medical treatment is a common issue in medicine, especially with chronic diseases such as arterial hypertension and diabetes, where omission of medication can result in clinically significant symptoms [3]. Nonadherence to therapy has also been described in many chronic eye conditions that require frequent or long term eyedrop administration, including in the treatment of corneal ulcers [4,5] and dry eye syndrome secondary to chronic ocular surface diseases or to refractive surgery [6,7] in infants, who may be uncooperative [8]. Furthermore, there is literature that indicates that another recognized barrier to the patient’s compliance is the cost of medication, as in the case of intravitreal injections, which are used for a variety of retinal conditions [9]. Given the asymptomatic nature of glaucoma and the lifelong therapeutic regimen without apparent subjective improvement, glaucoma patients are at risk of non-compliance with their treatment [10].

An ideal treatment regimen should achieve glaucoma control with the lowest risk and fewest adverse effects, whilst preserving the patient’s visual function and quality of life. Visual impairment due to glaucoma has a negative effect on physical and mental health. It seems that the simple knowledge of having a chronic, irreversible, and potentially blinding disease can have a negative impact on the quality of life of patients. The Collaborative Initial Glaucoma Treatment Study assessed the fear of blindness in patients with glaucoma through questionnaires. At the baseline, after being told about their glaucoma diagnosis, 34% of patients reported either a moderate or high degree of worry regarding blindness, which decreased to 17% by 6 months and to 11% over a 5-year period. The decrease noted over time was probably due to the reassurance associated with receiving treatment, regular clinical follow-up, adaptation to the diagnosis, and a good adherence to treatment [11]. Many studies found that glaucoma therapy adherence was associated with an increased quality of life. Although the quality of life may be a highly subjective concept, it should be regarded as a central aspect of therapeutic decisions in glaucoma as it can allow clinicians to understand patients’ perspective better and promote adherence to the treatment regimen [12].

Ophthalmologists are beginning to discover that nonadherence is more prevalent than previously realized and to understand the burden of glaucoma patients. Several studies of adherence with glaucoma therapy regimens point out that adherence on average is suboptimal, but research is still needed regarding how to detect and address nonadherence. The aim of this review is to provide an overview of patients’ compliance with glaucoma treatments. Perhaps physicians should consider patient-centred approaches such as empowering patients in their own treatment through education and support, which could be regarded as important first steps in reinforcing adherence or correcting nonadherence [13].

## 2. Materials and Methods

We conducted a literature search for this review using the MEDLINE database. The following search terms were included: “adherence to glaucoma medication”, “methods of measuring adherence”, “nonadherence”, “barriers to adherence with glaucoma medication”, “electronic medication monitoring”, “self-tonometry”, “topical treatment in glaucoma” and “questionnaires for glaucoma adherence”. The articles retrieved were reviewed for their title, abstract, and language (non-English articles were excluded). Inclusion criteria consisted of manuscripts written in English that had appeared before February 2022. We concentrated on the studies that best described patient adherence to a medication regimen, evaluated efficacy and/or compared methods regarding measuring adherence and new studies addressing adherence in glaucoma. After relevant articles were retrieved using these keywords, a search was conducted through the reference lists of the chosen studies and additional papers were selected.

## 3. Literature Review

### 3.1. Definition of Adherence and Compliance

Before discussing methods and barriers of adherence to glaucoma therapy, it is essential to first define compliance and adherence as the terms are often used synonymously. Adherence refers to the prevalence of use of the initial medication at various time points and is founded upon patients’ understanding of their illness severity, their belief in the efficacy of a treatment, and in their ability to control their symptoms by using this treatment [12]. On the other hand, the term ”compliance” has been defined as the extent to which the patient is passively following the physician’s orders or recommendations and suggests that the therapy is not based on a therapeutic alliance between the patient and the physician. Both terms imperfectly describe the medication-taking behaviour, but many physicians prefer the term “adherence”, as “compliance” denotes a degree of passivity (on the patient’s participation in their therapeutic regimen). Regardless of which word is preferred, the full benefit of the many effective medications that are available will be achieved only if patients accurately follow the prescribed treatment regimen [14,15,16].

### 3.2. Patient Adherence to a Medication Regimen

Despite evidence indicating the therapeutic benefit of adhering to a prescribed regimen, many patients do not take medications as prescribed. Studies have shown six general patterns of medication taking among patients with chronic conditions: approximately one-sixth come close to perfect adherence to a regimen; one-sixth take nearly all doses, but with some timing irregularity; one-sixth miss an occasional single day’s dose and have some timing inconsistency; one-sixth take drug holidays 3 to 4 times per year, with occasional dose omissions; one-sixth have a drug holiday monthly or more often, with frequent omissions of doses; and one-sixth take few or no doses while giving the impression of good adherence [13].

Adherence with glaucoma medications is a fundamental problem in the care of glaucoma patients as 24–59% fail to receive the intended or full effect of the treatment [17]. The barriers to adherence in glaucoma are complex, and glaucoma patients vary widely in the way they take their topical glaucoma medications. Patients naturally want to please the doctor, which makes them reluctant to admit to nonadherence [13]. The typical glaucoma patients are older adults who have inherent difficulties taking any medications, the most commonly cited being trouble with manual dexterity and inadequate vision, which are reasons for relying on others to administer their drops [18].

Lacey et al. identified multiple obstacles to adherence with anti-glaucomatous therapy: lack of initial education about application techniques, forgetting drops, drop-scheduling, as well as problems with drop supply. Sight preservation appeared to be the major motivation for adherence and thus, it appeared that despite these issues, the majority continued to administer drops due to faith in drop efficacy’s preserving their sight or because they already had symptomatic visual loss [19].

Only a few studies have investigated how patients store their prescription drugs at home. One study concluded that more than half of older patients comply with general drug storage recommendations and more than half of the drugs requiring refrigeration were not stored accordingly [20]. To our knowledge, glaucoma medication does not require refrigeration (but we assume that it would have been a reason for a lower adherence to treatment).

The Glaucoma Adherence and Persistency Study most often identified the following factors as possible barriers to compliance: cost (55%), forgetfulness (32%), fear or denial (16%), lack of understanding about glaucoma (16%), and regimen complexity (15%). From the physicians’ own perspective, the most important barriers identified were: lack of patient motivation to use drops (50%), lack of patient understanding about glaucoma (41%), inability to communicate why compliance is important (15%), and limited time spent with patients in the context of a brief office encounter (12%). In addition, the study defined 3 groups of physicians on the basis of the attitudinal and behavioural variables: reactive (41%), sceptical (44%), and idealistic (16%) physicians. The reactives were least likely to detect and address nonadherence proactively. The sceptics were least likely to discuss glaucoma or the importance of treatment with the patient when initiating treatment, because they were less likely to believe that they could change medication adherence. The idealists believed in addressing adherence and reported behaviour consistent with that goal, such as discussing a drug’s mechanism of action, educating patients on how to use drops, and using telephone appointment reminders, which have been shown to improve adherence. They were more likely to seek to understand and follow concepts that were important to the patient when they suspected nonadherence [21].

Failure to take medication as prescribed increases the risk that patients will not receive the intended benefit, often leading to negative sequelae, such as worsening of disease and higher healthcare costs overall. Thus, understanding factors associated with maintaining one’s medication regimen is important to patients and doctors.

### 3.3. Methods of Measuring Adherence

Before engaging in efforts to change the patient’s attitude on glaucoma therapy, the clinician first needs to detect non-adherence (and to realize that the attitude needs to be changed). Patient adherence to medication regimens has been monitored since the time of Hippocrates and, to this day, measurement of patient adherence is essential if management of poor adherence is to be adressed efficiently. The methods available for measuring adherence can be divided into direct and indirect methods, each method having advantages and disadvantages. A gold standard method to measure adherence in patients with glaucoma is required but has not been established yet.

The direct methods of measuring adherence include directly observed therapy, which has the highest accuracy, and measurement of concentrations of a drug in blood. The direct methods require expensive assays, can be burdensome for the ophthalmologist, and are susceptible to distortion by the patient [16].

Indirect methods of measurement of adherence include collecting patient questionnaires or self-reports, asking the patient to keep a medication diary, asking the patient about how easy it is to apply the prescribed medication, assessing clinical response, ascertaining rates of refilling prescriptions, using electronic medication monitors, and assessing children’s or elderly patients’ adherence by asking the help of a caregiver. Verbal questioning of the patient, questionnaires, and patient diaries are methods that are simple and relatively easy to use, but they can be altered by the patient [16].

Patients do not want to be perceived as misbehaving by their physicians, so they may withhold information about non-adherence. A possible solution is to encourage the patient’s participation in a partnership between him or her and the physician, based on valuing the patient’s beliefs, concerns, and preferences. Thus, a good patient should be perceived as someone who works with the physician to overcome the inevitable concerns about medication and logistical barriers to adherence [22,23]. To accomplish this, ophthalmologists can use self-reported adherence measures such as glaucoma therapy adherence questionnaires, which assess whether and what the patient understands about her or his medical regimen and the confidence she or he has in medication adherence [23]. Self-reported measures may be the most cost-effective and simple way for providers to evaluate the rate of non-compliance and contribute to a better understanding of the obstacles to, and the motivations for adherence with glaucoma medication and to explore potential methods to improve adherence [19,24].

#### 3.3.1. 24 h Intraocular Pressure Curve

Single-day IOP measurements during office hours can only provide limited information of treatment efficacy as it is widely known that IOP fluctuates during the 24-h period. The most common method for evaluating glaucoma patients’ IOP is via a diurnal tension curve, which encompasses multiple IOP readings at different time points during office hours. Mansouri et al. suggested that night-time measurements in the habitual position are very important for the accuracy of the IOP profile since the measurements more closely align with an individual’s circadian rhythm and natural body positioning. This intervention shows the impact of adherence to glaucoma treatment by delineating an IOP profile on a 24-h period, and it may be used when inpatient and outpatient IOP measurements differ greatly. In addition, this method may provide a better understanding of the true IOP-lowering effect of treatment by revealing a typical fluctuation of IOP over the circadian cycle, leading to a consequent improvement in adherence and treatment outcomes and the possibility of individualized disease management [25].

#### 3.3.2. Questionnaires

Questionnaires are a useful resource that can expose poor adherence to glaucoma therapy by asking whether the patients know why they are taking their medication, whether they have experienced any side effects to their eye drops, or what their motivations to take the treatment are [16]. Lacey et al. published a qualitative research study based on a patient interview, aiming to identify motivational factors for adhering to glaucoma medication. The interview consisted of questions regarding therapy, memory and routine of taking the eye drops, difficulties experienced with medication, motivations in applying the eye drops, and the patients’ ideas to help future adherence. The authors reported barriers to adherence derived directly from participants’ experiences, such as the lack of patient instruction and a desire for improved delivery of education, concentrating on drop application techniques and the consequences of poor adherence. Additionally, patients proposed the idea of receiving regular feedback about drop efficacy to strengthen their faith in adherence. Furthermore, the questions on adherence revealed that many patients experienced problems with drop application methods, forgetting drops, or practical difficulties such as being untreated for short periods due to running out of medication, finding a convenient location to apply the drops during the day, or having inadequate time to administer them while at work [19].

This method, however, is susceptible to error with increases in time between visits. Results are easily distorted by the patient, which usually leads to an overestimation of the patient’s adherence [16] A few studies evaluated the validity of self-reported measures against more controllable measures such as electronic monitoring devices or pharmacy records in glaucoma patients, and discovered that they tended to overvalue the doses taken and timing (adherence to glaucoma therapy) [24]. Consequently, Sayner et al. advanced the idea that ophthalmologists may need to use a careful and reassuring approach when talking to patients in order to detect pitfalls with applying drops on time, making them feel comfortable, and consequently helping the identification of poor adherence (for example: “I know it must be difficult to take your medication regularly. Tell me about the last time you forgot to take your drops” [16,24].

#### 3.3.3. Rate of Refilling Prescriptions

Rates of refilling prescriptions, as indirect methods of determining adherence, are an accurate measure of adherence in a medical system that uses electronic medical records and a closed pharmacy system as long as the refills are measured occasionally. Pharmacy records can provide the clinician with objective information on rates of refilling prescriptions, which can be used to assess whether a patient is adhering to the glaucoma therapy. Moreover, the records can be confirmed with the patient’s responses to direct questions or questionnaires [16]. However, fulfilment of a prescription at the pharmacy does not mean that the patient will apply the medication. The Glaucoma Adherence and Persistency Study analyzed large pharmacy databases to quantify adherence in a cohort of patients with glaucoma. The study highlighted several limitations of using pharmacy records to determine adherence to glaucoma medication. First, it showed that misclassification of added versus switched medication may have occurred, because patients mistakenly believed that the second medication should replace the first or a refill of the first drug was delayed because the patient had a large supply of it. In addition, patients who receive samples will be considered to have poor adherence. The samples given by the ophthalmologist do not appear in the claims data and this implies a difficulty in quantifying them, as well as in identifying the patients who may experience adverse effects or other barriers to adherence while taking the sample drug [26].

#### 3.3.4. Electronic Medication Monitoring

The most accurate indirect method of determining how patients use glaucoma medication is electronic monitoring with a Medication Event Monitoring System (MEMS), which is capable of recording and stamping the time of opening bottles and dispensing drops [16]. This automatic compilation of times of medication intake (dosing history) is an approach widely used in other medical specialities. It provides a thorough characterization of medication adherence, with clear distinctions between initiation, implementation, and discontinuation.

However, electronic monitoring of drop-taking is performed rarely in ophthalmic research, in part because of the technological difficulties involved. On the one hand, it implies a greater difficulty in measuring eyedrop usage compared with measuring pill usage, and on the other hand, alteration of the eyedrop bottle itself is expensive and difficult to achieve [27]. Therefore, an important drawback is the cost, which makes them impractical for monitoring adherence in clinical settings. Additionally, these devices do not document whether the patient actually applied the drop in the conjunctival sac. Thus, patients may open the container but not apply the drop correctly or may waste multiple drops out of the container at the same time [16]. Reports based on pharmacy records concluded that adherence and persistence (duration of continuous treatment with the initially prescribed medication) are higher for prostaglandins than with other drugs. Nevertheless, there has been no direct comparison using electronic monitoring and it will be of great interest to develop a method to compare adherence by electronic monitoring between the once-daily hypotensive agents and other glaucoma drugs that require more frequent dosage [27]. Another limitation of electronic monitoring of glaucoma medication is that patients who know they are monitored may change their behaviour simply because they are observed, a phenomenon called the Hawthorne effect [28].

A disadvantage is that electronic monitoring can only be used with specific medications, for example the Travatan Dosing Aid (DA; Alcon, Fort Worth, TX), which can only provide data on use of travoprost, because no other bottle of glaucoma medication fits within it. A bottle of travoprost is placed in the device and a handle is fully depressed to administer the medication. A built-in memory chip records the date and time of the event and the data is later downloaded to a computer [29].

The Travatan Dosing Aid Study assessed patients’ adherence and patterns of usage of topical once-daily therapy with travoprost (for glaucoma) in 196 patients over a 3 month period. The study revealed that, even though they were aware that they were being monitored and that they were provided with free medication, 45% of patients used their drops less than 75% of the time, thus displaying low compliance. In addition, patients reported considerately higher medication use than in reality and those with adherence of less than 50% of expected doses showed far higher dose taking immediately after the office visit and just before the return visit at 3 months. Finally, the study showed the poor ability of the physician to identify non-adherent patients, based on their self-reports, IOP measurements, or other subjective indicators. It concluded that in order to identify the less adherent patients, there is a need for better communication skills, better electronic monitoring, or both [27]. Robin et al. used electronic monitoring to objectively measure patient adherence with once-daily prostaglandin analogues as the sole ocular hypotensive therapy and to compare them with adjunctive medicine to the prostaglandin analogues and concluded that more complex dosing regimens result in poorer adherence, while once-daily drugs in a complex dosing regimen were found to have good adherence [30].

However, the studies we found advanced the idea that electronic monitoring of glaucoma therapy provides the most accurate data on adherence to glaucoma medication and very detailed knowledge regarding patterns of medication-taking behaviour.

### 3.4. Strategies for Addressing Nonadherence

The interventions to improve non-adherence represent an important goal of glaucoma research and require a complex approach, which is dependent on the patients’ needs and lifestyle [31]. Glaucoma is a unique chronic disease since the rates of adherence and compliance in the treatment of the condition are relatively low compared to other chronic conditions which require lifelong therapeutic interventions [32]. Currently, there is a paucity of research examining intervention strategies to enhance glaucoma medication adherence. Methods that can be used to enhance adherence in glaucoma therapy can be grouped into: patient education, improved communication between physicians and patients, simplifying and optimizing treatment regimens, and better patient interaction with the health care system. Budenz published a guide of interventions for improvement of glaucoma therapy adherence based on studies of the treatment of systemic hypertension, which promotes involving or empowering patients in the use of their own treatment and a proactive approach of the physician. Reviews of hypertension adherence studies have shown that three of the most straightforward strategies that improve adherence to therapy are to simplify treatment regimens, to optimize treatments to reduce side effects, and to reduce medication costs [16].

#### 3.4.1. Patient Education

The first step in improving patient adherence to a medication regimen is patient education (as glaucoma medications are effective only if patients use them) [1]. Education should be delivered in the form of both verbal and written instructions, including color-coded medication schedules, pictures, and adapted information for those with poor vision or low literacy. The ophthalmologist must make sure that the patient understands the treatment regimen and for this purpose, didactic presentations on proper drop application are important and should be repeated periodically to ensure that the patient continues to use the proper technique. Patients should be instructed about correct dosing schedules, minimization of waste of medication, and a clear discussion regarding the fact that vision can be irreversibly lost if the medications are not properly applied. Moreover, regular review of drug administration at each visit helps reinforce adherence by providing the physician with information regarding the patient’s level of understanding of his treatment regimen. In addition, patients should be encouraged to participate in their own treatment by keeping a daily record of medication dosage. With the availability of cell phones and Internet communication, there are several potential avenues that deserve exploration to improve compliance using continuous reminder systems, such as those available on the telephone (alarms, applications, text-messages, and phone calls) or e-mails, to reduce forgetfulness [16,33].

Situational and environmental factors play a role in adherence of glaucoma patients as there are individual specific daily situations that make it difficult for patients to be compliant [3]. It is very important to incorporate a regimen that is easy to use and easy to integrate into the daily lives of patients. Treatment adherence has been shown to be lower in the younger age groups, in spite of having greater access to electronic devices, often due to a busy lifestyle and work commitments. Therefore, involving patients can take the form of asking them to integrate their treatment into their daily activities. For example, they can take a bottle of drops at work or put the drops next to the bed, desk, or toothbrush. Moreover, physicians may suggest involving a helpful family member to assist with applying drops or reminding the patient to take drops [16]. Even though young patients can manage online information without the physician’s advice, it does not seem to improve their adherence. There is evidence showing that adherence with glaucoma treatment is lower in the first year of diagnosis, and improved education about the disease may help patients persevere with their treatment [34].

#### 3.4.2. Patient-Physician Communication

Carpenter et al. studied whether patient–physician communication increases the medication self-efficacy of glaucoma patients and concluded that it can improve the treatment adherence. Thus, providers should adopt a patient-centred approach, taking time to educate patients about their chronic disease and to assess their views on the treatment. The study also indicated that patients who ask more medication-related questions may have less confidence that they can adhere to their glaucoma regimen and should receive more support to address the adherence barriers [2]. A study of 279 patients with glaucoma who were video-recorded during their visit showed that educating patients about their condition occurred during approximately two-thirds of the visits, but was not significantly associated with whether patients took their doses on time during the next period of time after the visit. Instead, education regarding administration of the eye drops was the only provider communication variable that was significantly associated with adherence. Accordingly, thorough understanding of glaucoma must be supported by health professionals by allocating more time to providing information to patients [35].

An observational study investigated an intervention program consisting of education and a reminder system. The adherence rate with glaucoma medication increased from 54% to 73% in patients whose baseline drop-taking was under 75%. Improvement was immediate and sustained over 3 months. The research suggested that using a multifaceted approach improved the probability that the interventions changed medication use. The study could not determine which aspects of the intervention were most valuable and which strategies can be implemented in clinical practice. Additionally, the researchers found that there was greater improvement in adherence among white patients and those with the lowest baseline adherence. Additionally, improvement in adherence did not correlate with the level of IOP measured in the clinic. To this end, the results of the study showed that many nonadherent patients had satisfactory IOP at routine visits and thus, measurement of IOP was considered inadequate for estimating adherence. Finally, it was underlined that further research is needed to distinguish poor adherence in patients to avoid overmedication and to determine which elements of the adherence program were most effective [33].

#### 3.4.3. Simplifying Medication Regimen

Interventions designed to increase patients’ medication-related self-efficacy should consider prescribing simpler medication regimens. The importance of simplifying patients’ medication regimen has been emphasized by many studies that showed that patients on more complex regimens were less likely to take their doses on time and they were less likely to take the correct number of prescribed doses each day [30,35]. Additionally, it was demonstrated that patients taking once-daily drugs in a complex dosing regimen were found to have good adherence [30]. It was shown that 3 and 4 times-a-day dosing and improper spacing of doses, versus twice-a-day dosing, increase noncompliance; hence, the fewest number of medications, instilled with the least frequency, enhance patient satisfaction and dosing convenience [36].

Shirai et al. evaluated Japanese patients’ adherence to fixed and unfixed combination eyedrops and found that patients with fixed combination therapy had a higher adherence to their regimen. The main reason may be that a fixed combination is more convenient for patients. It is easier and faster to apply one drop of an ophthalmic solution than two drops from separate bottles of medication, with no waiting period between the two types of drops, with similar or possibly superior efficacies in daily practice compared with unfixed-combination therapies. Furthermore, it is more advantageous because of the decreased exposure to preservatives and thus decreased risk of developing ocular surface diseases [37].

Possible solutions when patients require multiple medications and doses would be coordinating administration with daily events, such as meals or brushing teeth and using a written schedule for medications [1]. Furthermore, patients’ compliance is enhanced when patients are aware of the possible adverse effects of a medication and patient education should include a discussion of treatment alternatives. Research has suggested that individuals receiving preservative-free medication demonstrate lower rates of nonadherence (12.5%) and it has been indicated that switching to preservative-free therapies may be of particular benefit regarding adherence with self-administered treatments [38].

#### 3.4.4. Medication Cost

Many studies have confirmed that the majority of glaucoma office visits do not include a discussion of medication cost. Providers often do not ask if their patients have glaucoma medication cost problems and, in turn, although patients may experience financial dificulties with regard to the treatment regimen, they rarely mention it to their doctors. Therefore, ophthalmologists should consider mentioning medication cost during office visits in order to improve adherence from the beginning of glaucoma treatment [39]. A possible solution to overcome the barriers of cost and the physical burden of drug acquisition would be to provide free eye drops to patients, either by occasionally giving free samples or by rendering the medication reimbursable.

Finally, a cost-utility analysis assessed the societal costs of optimal versus poor adherence to glaucoma medications among people over 40 years of age with newly diagnosed glaucoma on a 60-year time horizon and demonstrated that being adherent to glaucoma medications resulted in improved quality of life for a relatively low increase in lifetime healthcare costs [40].

#### 3.4.5. Self-Measurement of Intraocular Pressure

The measurement of intraocular pressure (IOP) is one of the most important diagnostic methods to determine the effectiveness of glaucoma treatment. However, a single measurement of IOP during office visits does not reveal the true level of intraocular hypertension and of the daily fluctuations in IOP. Astakhov et al. assessed the convenience of patients’ self-monitoring their IOP at home using Icare^®^ HOME tonometers and noted an increase in the level of adherence to treatment. The study showed that independent participation of the patient in the diagnostic process improves awareness and understanding of the disease and the importance of following the doctor’s instructions [41].

Chen et al. evaluated the reliability of measurements made by patients themselves using iCare tonometers and compared these with Goldmann applanation tonometry measurements. The study showed that patients appreciated the method of self-measuring their IOP at home as it has the advantage of saving time for both the patients themselves and the glaucoma care providers. The iCare One self-tonometer allowed patients to instantly read the results, which made them feel positively toward this possibility. Moreover, patients wished this method could be a part of future glaucoma monitoring. Thus, home IOP monitoring provides valuable information for glaucoma care and might improve adherence to treatment [42].

## 4. Conclusions

In summary, the body of literature on glaucoma medication adherence shows that it is a challenging problem for ophthalmologists who provide services to patients with this chronic disease and the many tested interventions have had varying degrees of success.

We believe ophthalmologists should suspect a potentially reduced adherence when the IOP level is not congruent with the prescribed regimen, when the patient has not been well educated and does not understand his chronic disease and the importance of treatment on his vision-related outcome, or when the patient has a complicated medication regimen. Additionally, we must take into consideration that most glaucoma patients are older adults who may have physical or cognitive disabilities and limited financial resources as well as impaired visual acuity, which may interfere with the administration of eyedrops. Daily routine measures to detect reduced adherence, which are simple and relatively easy to use, include evaluation of patient diaries and questioning the patient about difficulties encountered with applying the treatment, drops schedule, side effects of the eyedrops, and the patient’s opinion regarding the regimen.

Measures to improve adherence, which may have a great effect on boosting the patient’s ability to follow a medication regimen are patient education, therapy reminder systems (e.g., alarms, text-messages), simplifying the therapy and tailoring it to the individual’s lifestyle with patient participation, preservative-free medication to lower the possible side effects, and better communication between the patient and the health care provider. Integrating a person-centred method may help in treating the patient as a whole by allowing a focus on both the eye disease and quality of life.

Even though there are myriad causes of nonadherence and there remain large gaps in detecting, identifying, and addressing nonadherence, a multifaceted approach seems to be a problem-solving strategy [13,16].

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
