# Peer review of "Adherence to Therapy in Glaucoma Treatment—A Review"

_jpm, 2022, doi:10.3390/jpm12040514_

Round 1

Reviewer 1 Report

The manuscript is overall well written; the examined topic is, as the authors present, of great clinical relevance. I find it interesting and relevant, that you discussed the patient as well as the physicians point of view on this subject.

I would still recommend some improvements to enhance readability and comprehensiveness.

Major comments:

It would be nice to have the difference between adherence and compliance explained/ have these terms defined as they are not exact synonyms. Also the use of these terms throughout the manuscript should be reevaluated accordingly.

The structure of the section 3.2 “methods of measuring adherence” could be made more visible by using subsection titles like “questionnaires”, “Rate of prescription refilling”, “Electronic medication monitoring”, etc; and the structure of the discussion within these subsections could be improved, in particular to avoid redundancy.

The structure and readability of the 3.3 section “Strategies for addressing nonadherence” could be improved similarly.

Another intervention that can give information on therapy adherence and a patient’s ability to administer eye drops is their admission for a 24-hour intraocular pressure curve (e.g., if inpatient and outpatient IOP measurements differ greatly). This could be added to the manuscript.

The conclusion should add more precise recommendation based on the authors present analysis of the literature und personal suggestion regarding 1) when ophthalmologists should think about possible reduced adherence (IOP out of target?, when patients don’t know which drops are to be taken?), 2) daily routine measures to detect reduced adherence (e.g., ask patients about difficulties, side-effects?), 3) measures to improve adherence (patient education, therapy reminder, simplification of therapy, side-effect reduction?).

Minor comments:

Line 27: “this goal”: it is not very clear what this refers to.

Line 28: “disruption to the patient’s life”: maybe talk about quality of life?

Adherence is also an important aspect when determining a patient’s individual target IOP, since it is related to their quality of life (cf. Guidelines of the European Glaucoma Society). This paradox relation between therapy adherence and quality of life could also be discussed, as it is a central aspect of therapeutic decisions in glaucoma.

Line 184: “errors from sampling” can be misunderstood as sampling error from a statistical aspect; this should be rephrased.

Some language improvements are needed, e.g.:

Line 30-31: “lapses in therapy” …

Line 34: “who 34 are often difficult to collaborate with”

Line 42: “and the challenges faced by glaucoma patients

Line 50-51; line 83-86;

Line 89-90: “in classifying”…

Line 113-114: “until today,”…

Line 127-128, 130-132, 173-175

Line 197: “in part in part”

Line 216-217: “no other recipient”…

Line 228-231

Final general suggestions:

Here are some ideas that could be worth discussing in your manuscript (suggestions as I’m not sure about how this aspects were already examined/ published):

Effect of fixed combination of IOP-lowering agents (e.g., dorzolamid and timolol).

Do medication to be stored cool reduce adherence compared to medications stored at room temperature?

What about the role of self-/ home-tonometry as a method to improve the patients awareness and understanding of the disease and making the efficacy of the therapy more visible, all to improve adherence.

Effect of unconventional working hours (night-shifts) and alternating working hours (day and night-shifts) on adherence.

Reviewer 2 Report

  1. A bit concerned about the search method - only one database was used, and the description of the search term was not very clear. Will be better if a more rigorous search strategy was used.
  2. Does the study adhere to certain guidelines for systematic review (e.g. PRISMA)? Although the title said "systematic review", the current study is more like a narrative review.
  3. Adding some more subheadings may further improve the reading experience. E.g. 3.2. Methods of Measuring Adherence -  "3.2.1 Questionnaire", "3.2.2 Rates of refilling prescription", "3.2.3 Electronic monitoring, etc.
  4. While there is not significant problem with the writing skills, a second check for grammatical mistake and typo is still suggested.

E.g. Introduction - "dry eye syndrome associated with"

Introduction - "in infants, who are often difficult to collaborate with" - is there any missing words in the sentence? This sentence does not seem complete.

Round 2

Reviewer 1 Report

The authors addressed my comments comprehensively.

Here are a precision on the to comments I formulated to imprecisely in the first review round (sorry for that!); the following phrases could be rephrased or shorten slightly to improve readability:

From line 281 and following: “Finally, the study illustrated…”

From line 227: “In this case, pharmacy reports…”

Regarding persons working night-shifts, what I ment was that changing from day to night-shift can render the observation of the eye drop application rhythm difficult, but I didn't find any paper on this topic either... It would be interesting to investigate... 

Congratulations on your thorough work overall!
